# Mitochondrial DNA on Tumor-Associated Macrophages Polarization and Immunity

**DOI:** 10.3390/cancers14061452

**Published:** 2022-03-11

**Authors:** Yaxin Guo, Hsiang-i Tsai, Lirong Zhang, Haitao Zhu

**Affiliations:** 1Department of Medical Imaging, The Affiliated Hospital of Jiangsu University, Zhenjiang 212001, China; gyx961225@163.com; 2Laboratory of Radiology, The Affiliated Hospital of Jiangsu University, Zhenjiang 212001, China; tsaihsiangi88@163.com

**Keywords:** mitochondrial DNA, macrophage, innate immunity, macrophage activation, macrophage biology, cancer

## Abstract

**Simple Summary:**

As the most abundant cell in the tumor microenvironment (TME), tumor-associated macrophages (TAMs) drive tumor progress by inducing angiogenesis, fibrosis, invasion, metastasis, and immunosuppression, which makes these cells an important target for tumor treatment. Recently, the role of free mitochondrial DNA (mtDNA) has attracted increased attention in the regulation of immune cells in the TME. In this review, we first summarize the functional characteristics of macrophages in tumor progression. The release and regulation mechanisms of tumor cell-derived mtDNA in TME are also introduced. Then, the biological effects of endogenous and exogenous mtDNA on macrophages are discussed. Finally, we propose that the effect of mtDNA on macrophages is worthy of attention in the process of tumor treatment, especially in immunotherapy. Our review provides a systematic summary of the effects of mtDNA on the survival, function, and phenotypes of TAMs in the TME.

**Abstract:**

As the richest immune cells in most tumor microenvironments (TMEs), tumor-associated macrophages (TAMs) play an important role in tumor development and treatment sensitivity. The phenotypes and functions of TAMs vary according to their sources and tumor progression. Different TAM phenotypes display distinct behaviors in terms of tumor immunity and are regulated by intracellular and exogenous molecules. Additionally, dysfunctional and oxidatively stressed mitochondrial-derived mitochondrial DNA (mtDNA) plays an important role in remodeling the phenotypes and functions of TAMs. This article reviews the interactions between mtDNA and TAMs in the TME and further discusses the influence of their performance on tumor genesis and development.

## 1. Introduction

In response to harmful stimulation and tissue damage, macrophages are involved in tissue homeostasis and immune defense against pathogens by removing dead cells and foreign substances [1]. Tumor-associated macrophages (TAMs) are abundant in most solid tumors and facilitate tumor progression, including promoting genetic instability and neovascularization, thereby inducing fibrosis, invasion, and metastasis [2,3]. TAMs also create an immunosuppressive microenvironment by triggering inhibitory immune checkpoints in T cells, thus promoting fibrosis and lymphocyte rejection [4]. Therefore, TAMs are considered to be associated with a poor prognosis.

TAMs have high plasticity and are roughly divided into two categories: classic macrophages (M1), which produce pro-inflammatory cytokines, and alternatively activated macrophages (M2), which produce anti-inflammatory cytokines. In addition, CD169^+^ and TCR^+^ TAMs represent a newly identified subtype [5]. The phenotypes and functions of TAMs are highly heterogeneous and closely regulated by tumor processes as well as many small molecules in the TME, including various cytokines, interleukins, nucleic acid, and other pathogen-associated molecular patterns (PAMPs), in addition to damage-associated molecular patterns (DAMPs), mitochondrial metabolism, and lactic acid [6]. Mitochondrial DNA (mtDNA) is an important DAMP released by damaged mitochondria after infection and stress and can induce inflammatory responses through a variety of pattern recognition receptors (PRRs) [7].

Mutation and dysfunction of mtDNA, an important genetic component of mitochondria, can be used to assess the occurrence and development risk factor of cancer [8,9]. Compared with genomic DNA, mtDNA is located in the oxidative and high-fat environment of the mitochondria. When mitochondrial oxidative phosphorylation (OXPHOS) occurs, mtDNA is directly exposed to the accumulation of toxic reactive oxygen species (ROS) byproducts, which makes mtDNA more prone to oxidative damage [10]. In addition, due to the single-stranded D-loop of mtDNA, tRNA is prone to a hairpin-like structure, which increases the chance of mismatching [11]. Furthermore, because mtDNA lacks chromatin protective structures such as histones and introns, has low repair efficiency, and offers few repair pathways, mtDNA is more susceptible to oxidative stress damage, such as ROS [12]. Free mtDNA is released into the cytoplasm and extracellular matrix as a DAMP and is further involved in monocyte-mediated anti-tumor immunotherapy [13,14].

Free mtDNA was previously thought to be an endogenous tumor antigenic substance. The true regulation of mtDNA in terms of the number, function, and phenotypes of macrophages has recently attracted increasingly more attention. In this review, we discuss how various derived free mtDNA accumulate in macrophages; regulate macrophage recruitment, survival, polarization, and function; and ultimately determine the immunity of the tumor microenvironment.

## 2. TAMs in Tumor Progression

Solid tumors are rich in TAMs [15]. In certain solid tumors such as breast cancer, TAMs can constitute up to 50% of the tumor mass [15,16]. In most tumors such as glioblastoma, ovarian cancer, colorectal cancer, lung cancer, and prostate cancer, the content of TAM is not only high but also closely related to tumor growth, metastasis, and treatment sensitivity [17,18,19]. Because of the strong plasticity of macrophages, infiltrated TAMs are a “double-edged sword” in the occurrence and development of tumors. Under the actions of different stimulating factors, macrophages are divided into the M1 subtype or M2 subtype according to their phenotypes and functions [20,21].

M1 macrophages often appear in the early stages of tumors to monitor transformed cells and play an anti-tumor effect. With the accumulation of lactate in the TME and the expression of vascular endothelial growth factor (VEGF) and Arginase-1 (Arg-1) in macrophages, infiltrating macrophages are induced to switch to the M2 subtype [22,23,24]. Under the stimulation of Interferon-γ (IFN-γ), lipopolysaccharide (LPS), granulocyte-macrophage colony-stimulating factor (GM-CSF), and other substances, macrophages are polarized to the M1 subtype [25,26]. To present antigens and activate adaptive immune responses, M1 macrophages often express high levels of major histocompatibility complex II (MHC-II) molecules and costimulatory molecules such as CD40, CD80, and CD86 [27]. The M1 macrophage directly produces nitric oxide (NO), ROS, and reactive nitrogen species to inhibit tumor growth [28]. Moreover, M1 macrophages enhance helper T (Th1) adaptive immune responses by enhancing the antigen presentation of the MHC complex and secreting various pro-inflammatory factors such as tumor necrosis factor-α (TNF-α), Interleukin-1 alpha (IL-1α), IL-1β, IL-6, IL-12, IL-18, and IL-23 to promote inflammation, thereby helping to eliminate tumor cells [28]. Macrophages similar to the M1 subtype usually infiltrate more frequently in the early stages of tumors, which can allow cancer cells to play an anti-tumor effect and the released pro-inflammatory factors to exert and amplify anti-microbial and tumor-killing activities, as well as improve the Th1 adaptive immune response by simultaneously enhancing antigen presentation. However, recent studies also discovered that M1 macrophages induce local chronic inflammation in the early stages of tumor formation and promote tumor progression [29,30], possibly due to the greater dependence of early tumors on environmental stimuli.

In most solid cancers, such as breast cancer, pancreatic cancer, and bladder cancer, an increase in TAM infiltration has long been associated with a poor prognosis, which highlights the value of TAMs as potential biomarkers for cancer diagnosis and prognosis [31,32]. In tumor progression, the activation state of TAM is similar to that of M2-like macrophages (also known as alternatively activated macrophages), which express distinct markers, including mannose receptor type c 1 (CD206) and scavenger receptor CD163. Notably, M2-like macrophages can be further subdivided into M2a, M2b, M2c, and M2d according to their gene expression profiles. The M2a subtype is involved in tissue repair and healing following activation by IL-4 and IL-13 [33]. Once activated by the immune complex and LPS, the M2b subtype promotes IL-10 production and plays an anti-inflammatory and immunomodulatory role [34]. The M2c subtype, which is induced by IL-10, transforming growth factor-β (TGF-β), and glucocorticoid, participates in immunosuppression and tissue remodeling [35,36]. The M2d subtype, which is mainly activated by tumor-related factors such as IL-6, promotes tumor growth and angiogenesis, which is the main component of TAMs and promotes tumor growth (Table 1) [37]. Therefore, all M2 subtypes can produce anti-inflammatory cytokines. 

Under the stimulation of cytokines, such as colony-stimulating factor-1 (CSF-1), IL-4, IL-10, TGF- β, and IL-13, M2 subsets are activated and secrete inflammatory inhibitory factors such as IL-10, Arg-1, chitinase-like protein (Ym1), and VEGF, which are closely related to tumor growth, development, invasion, and metastasis [62]. Moreover, M2 macrophages can directly or indirectly inhibit the adaptive immune cell response. (1) M2 macrophages inhibit the function of killer cells. TAM-derived TNF-α upregulates the expression of programmed death 1 ligand 1 (PDL-1) on tumor cells, thereby inhibiting the function of T cells, natural killer (NK) cells, and activated dendritic cells (DCs) [63]. Meanwhile, M2 macrophages can promote the up-regulation of PDL-1 expression through autocrine VEGF signaling, inhibit CD4^+^/CD8^+^T cells, and increase CD4^+^CD25^+^regulatory T cells (Tregs) in the bone marrow [64]. Furthermore, the TGF-β and IL-10 secreted by macrophages convert Th1 cells into Th2 cells, thereby reversing the antitumor effects of CD8^+^ cytotoxic T cells and CD4^+^ Th1 cells [65]. Therefore, M2-subtype TAMs can facilitate tumor immune escape by inhibiting adaptive immune functions. (2) M2 macrophages, moreover, recruit immunosuppressive populations. It was previously reported that M2 macrophages can induce Treg infiltration into the tumor microenvironment, inhibit T-cell immunity, and promote tumor growth by overexpressing CCL1 or CCL22 [66,67]. 

Simultaneously, TAMs can promote the proliferation and activation of Treg by secreting IL-23 and inhibit cytotoxic T lymphocytes (CTLs) from killing tumor cells [68]. TAMs can also further increase the infiltration of myeloid-derived suppressor cells (MDSCs) by expressing CD39 and CD73 [69]. N2-type tumor-associated neutrophils (TANs) are also a classic immunosuppressive cell, and there is evidence that TAMs are involved in the recruitment of N2-type TANs and their function in promoting tumor metastasis. It appears that M2-type TAMs can promote tumor immune evasion by affecting various immune cell functions [70]. (3) TAMs can also indirectly promote the proliferation of blood vessels in the tumor and regulate the environment of the extracellular matrix (ECM) and chemokine to promote tumor growth [71,72,73]. Therefore, inhibiting the M2 subtype or inducing TAM to yield M1-subtype polarization is a promising therapeutic avenue. 

## 3. Regulation of Free mtDNA in TME

Mitochondrial genetic material mtDNA, an endogenous nucleic acid, is widely spread in organisms and has a dual role in the occurrence and development of tumors. When the tumor develops oxidative stress after treatment, mtDNA is damaged and released through the mitochondrial membrane, thereby forming in situ DAMPs, which may have important research value in the regulation of TAM functions and the remodeling of TME after radiotherapy and chemotherapy. The mechanism of mtDNA accumulation in TME is discussed in the following section.

### 3.1. Release of Tumor-Derived mtDNA

In the resting state, many unique nucleic acid species are generated due to the special D-Loop structure of mtDNA and the frequent suspension or termination of mtDNA replication, including long, double-stranded RNA, uncapped mRNAs, and RNA–DNA hybrids [74]. MtDNA damage can also be caused by DNA replication errors and DNA mutations [75]. In addition, due to the special metabolic characteristics of tumors, a large amount of lactic acid accumulates in mitochondria, mtDNA, which is more unstable and easily damaged than normal cells [76]. The mtDNA in TME is mainly derived from oxidative stress-damaged cancer cells. Various types of oxidative stress, such as hypoxia, ionizing radiation, DNA damage-induced drugs, and other pathological conditions, trigger mitochondrial damage and further produce mitochondrial reactive oxygen species (mROS) [77]. ROS can react with mtDNA to induce the oxidation of guanine to 8-hydroxyguanosine (8-OHG) resulting in stress and breakage of mtDNA due to oxidization; mtDNA is ultimately released from the mitochondria to the cytoplasm in the form of oxidized mtDNA [78]. In addition, radiation therapy significantly induces activation of the ZBP1-RIPK3-MLKL necroptosis pathway, which is the dominant factor that promotes mitochondrial instability and releases mtDNA. Moreover, the damage and release of mtDNA can be amplified by inhibiting casepase8 [79]. Therefore, the mtDNA of tumor cells can become damaged, especially after conventional radiotherapy and chemotherapy. How the medical field can this characteristic to further amplify anti-tumor effects is worth exploring. 

The export of mtDNA mainly relies on non-specific pores in the mitochondrial inner membrane, voltage-dependent anion channel (VDAC) oligomers on the outer membrane, mitochondrial outer membrane permeability (MOMP), and mitochondrial permeability transition pores (MPTPs) [80,81]. Part of the mtDNA can be released in the form of mitochondrial-derived vesicles (MDVs). Kiichi Nakahira et al. previously found that pyrin domain containing 3 (NALP3) inflammasomes play a role in MPTP mediating mtDNA release [82]. The absence of NALP3 protects mitochondrial membrane potential and inhibits LPS and ATP-induced mtDNA release [82]. Bcl-2 homologous killer/Bcl-2-associated X protein (BAK/BAX) induces the permeability of the mitochondrial outer membrane and the appearance of MOMP in the outer membrane, which introduces mtDNA into the cytoplasm [83]. A. Gómez et al. revealed that the Gasdermin protein also regulates mtDNA release. Specifically, the Gasdermin protein was found to induce rupture of the mitochondrial membrane network and promote the release of mtDNA into the cytoplasm in the processes of pyroptosis and apoptosis. In addition, the knockout of gasdermin-D (GSDMD) was found to block the release of mtDNA into the cytoplasm in macrophages. Meanwhile, GSDMD enhances plasma membrane permeability and promotes mtDNA release [84]. The regulation of mtDNA release plays an important role in the accumulation of mtDNA in the cell and its immunological effects.

### 3.2. Regulation of Tumor-Derived mtDNA Accumulation

To reduce unnecessary inflammation, the release and accumulation of mtDNA in the cytoplasm can be prevented through several mechanisms, including (1) mtDNA self-repair. Some protective protein nuclear repair factors are present in the mitochondria, such as mitochondrial transcription factor A (TFAM), which is encoded by nuclear genes and forms a complex with mtDNA [85]. TFAM is uniformly distributed in the mitochondrial network in the form of nucleoids and is the main component involved in the replication, transcription, and repair of mtDNA [86]. TFAM can not only stimulate mtDNA transcription and regulate the quantity of mtDNA but also can protect mtDNA from oxidative damage and provide defense against oxidative products such as ROS [87]. Other repair pathways include mtDNA polymerase gamma (POLG), a repair factor involved in mtDNA synthesis, and AP endonuclease (APE1) protein, which offers DNA repair and redox functions [88,89]. These pathways can repair mtDNA and maintain mtDNA homeostasis in normal tissues. (2) The second mechanism is the reduction of mtDNA accumulation. A cascade of apoptotic caspases drives the rapid disintegration and clearance of apoptotic cells, which reduces the accumulation of mtDNA. Autophagy can transport damaged mitochondria, mtDNA, and other toxic substances into the lysosomes for degradation, which further affects the killing and abscopal responses of radiotherapy in breast cancer [90,91]. The ATM-CHK2 transduction protein is important for DNA damage repair responses (DDRs); this protein promotes autophagy to maintain cell homeostasis and inhibits innate immunity and lymphocyte infiltration [92,93]. MDVs transport damaged mtDNA to lysosomes and inhibit the accumulation of mtDNA in the cytoplasm. These avenues may represent a new adaptive mechanism that can enable cancer cells to compensate for autophagy loss and maintain their mitochondrial functions [94]. The imbalance between mtDNA repair and damage results in large amounts of mtDNA fragments releasing from the mitochondria and accumulating in the cytoplasm; these fragments can even be released into the extracellular environment (Figure 1).

## 4. Regulation of mtDNA in Immune Cells

MtDNA plays a dual role in regulating the TME. After recognizing the damaged mtDNA, DCs up-regulate CD86, CD83, and human leukocyte antigen (HLA-DQ) expression and enhance the transcription and release of TNF-α and other inflammatory factors, thereby contributing to forming an anti-tumor inflammatory microenvironment [95,96]. The damaged mtDNA swallowed by DCs can trigger a stimulator of interferon genes (STING)-dependent type I interferon response and enhance the antigen cross-presentation ability of DCs. At the same time, radiotherapy-induced damaged mtDNA promotes CD8^+^ T-cell proliferation and boosts these T-cells’ anti-tumor effects [78]. 

The interactions of TANs and mtDNA can affect the progression of tumors [97,98]. Several studies have reported that activated neutrophils expelled extracellular traps (NETs) containing mtDNA, which promoted the release of interferons and other pro-inflammatory cytokines to form a chronic inflammatory environment, which contributed to cancer cell growth and metastasis [99,100,101,102,103]. In addition, the mtDNA in TME can recruit and activate TANs and platelets [102]. It seems that the interactions between mtDNA and TAN can also promote tumor growth and metastasis. Therefore, mtDNA could promote both pro-inflammatory and anti-inflammatory TME according to different immune cells (Table 2). In the following section, the biological and anti-tumor effects of mtDNA on TAMs are detailed.

### 4.1. Exogenous mtDNA on Macrophages

The biological functions of macrophages are regulated by both endogenous and exogenous mtDNA. (1) Exogenous mtDNA can promote the recruitment of macrophages to the TME. In the mtDNA-free breast cancer cell line 4T1 (4T1ρ0), macrophage infiltration into tumor tissue was found to be significantly reduced in the early stages due to the lack of chemokine ligand 2 (CCL2), chemokine ligand 5 (CCL5), and the CXC chemokine ligand 10 (CXCL10) in 4T1ρ0 cells, which could be reversed in the presence of mtDNA [108]. Han Jiang et al. reported that a nano-catalytic drug (MRF) induced a Fenton reaction, which further promoted cancer cells mitochondrial damage and mtDNA leakage. The leakage of mtDNA in the extracellular matrix recruited macrophages to tumor tissues [109]. (2) Extracellular mtDNA promoted the polarization of macrophages to a pro-inflammatory phenotype with the up-regulation of CD86 and down-regulation of CD206 macrophages [109].

MtDNA can also indirectly affect the phenotypes and functions of macrophages by enhancing the release of cytokines. MtDNA fragments activate the TLR9 pathway and STING signal pathway to induce TNF-α release, which was reported to turn peripheral microglia into a proinflammatory phenotype in cerebral ischemic stroke [110]. Additionally, inhibiting the STING signal reduced the M1 polarization levels of macrophages [111]. TNF could up-regulate M1 pro-inflammatory gene expression by inhibiting the IL-13 secreted by eosinophils and down-regulating the expression of M2-related genes. In this way, the down-regulated transcription level of TNF or suppressed type I TNF receptor can significantly up-regulate the M2 macrophage-related gene transcription level [112]. 

MtDNA can also induce immunosuppressive M2 phenotype macrophages. MtDNA stress-activated TLR9 was also found to induce CCL2, IL-6, and IL-8 production in various solid tumors, thereby promoting macrophage infiltration and maintaining the immunosuppressive phenotype of macrophages [113,114,115,116]. In particular, IL-8 recruited MDSCs and induced tumor tissue-resident macrophages to polarize into M2 macrophages, which further inhibited the activity of cytotoxic T cells [117]. Meanwhile, IFN-I, which relies on the mtDNA activated STING signal, was reported to affect the polarization state of macrophages. Alan Bénard et al. found that in mouse and human mycobacterium tuberculosis infection models, B cells released mtDNA-STING-dependent IFN-I polarized macrophages towards an anti-inflammatory phenotype (Figure 2) [118]. The induction of distinct macrophage phenotypes through MtDNA results from cytokine release quantities and regulatory mechanisms. Therefore, the exact role of mtDNA in pro-inflammatory and anti-inflammatory TME is still controversial and needs to be further studied.

### 4.2. Endogenous mtDNA on Macrophages

#### 4.2.1. MtDNA Accumulation in Macrophages

MtDNA in macrophages can be sourced from phagocytes or be inherent. Oxidatively stressed mitochondria release damaged mtDNA into the extracellular following radiotherapy and chemotherapy. Extracellular mtDNA, as exogenous foreign antigens, can be engulfed by macrophages through endocytosis or MDVs [119,120,121]. By blocking the “don’t eat me” signal, antigenic surface determinant protein OA3 (CD47) can enhance the phagocytic functions of macrophages and promote mtDNA accumulation in macrophages [95]. In addition, inherent mtDNA is released into the cytoplasm under ionizing radiation, DNA damage drugs, oxidative stress, carcinogenic signals, telomere shortening, and chromosome separation [120]. 

TAMs also have protective mechanisms against external environmental stress that protect TAMs from mitochondrial damage and mtDNA-induced stress. For example, under the conditions of long-term lactic acidosis in solid tumors, mitochondria are damaged, and the mtDNA copy number is significantly reduced in macrophages. Meanwhile, the natural metabolite acetoacetate (AcAc) produced by the liver can be used by macrophages as an alternative fuel to protect the mitochondrial structure [76]. Mitochondrial-targeting antioxidant MitoQ prevents liver damage by scavenging mitochondrial ROS in kuffer cells (KCs), maintaining KCs’ mitochondrial integrity and functions and inhibiting mtDNA release [122]. In addition, the autophagy protein ATM-CHK2-Beclin1 contributes to the integrity and function of macrophage mitochondria in the presence of LPS, suppresses mtDNA accumulated in the cytoplasm, and prevents the secretion of IL-1 β and IL-18, thus inhibiting unnecessary inflammatory responses [82]. 

Excess mtDNA can be rapidly degraded in the cytoplasm of macrophages. The phagosomes and lysosomes effectively fuse, resulting in acidification of the phagosomal cavity and lysosomal deoxyribonuclease II (DNase II) activation, which further degrades mtDNA [123,124]. In addition, three initial repair exonuclease 1 (TREX1), also known as DNase III, located in the cytoplasm of phagocytes, can also degrade mtDNA [125]. Lacking DNase II or TREX1 was found to yield excess mtDNA accumulation in macrophages. Therefore, mtDNA content in the cytoplasm of macrophages results from a balance between mtDNA repair, degradation efficiency, and environmental stress damage.

#### 4.2.2. Impact of Instinct Endogenous mtDNA on Macrophage Biology

The impact of endogenous mtDNA on macrophage biology was mostly reported in non-tumor models. These effects included (1) regulating the death of macrophages. In an LPS-induced acute lung injury model, exogenous mtDNA stimulation increased the amount of mtDNA in the macrophage cytoplasm, triggering STING phosphorylation and macrophage pyroptosis, inflammation, and oxidative stress, which were reversed in the STING knockout model [126]. In the trauma tissue, endonuclease G induced mtDNA enzymatic hydrolysis and accumulation in the macrophages, which triggered macrophage necroptosis. Necrotic macrophages further triggered the surrounding naive macrophages to polarize to the pro-inflammatory phenotype [127]. MtDNA also (2) affects the function of macrophages. MtDNA in acute myeloid leukemia (AML) apoptotic cells can induce the activation of STING in bone marrow macrophages, promote macrophage LC3-associated phagocytosis (LAP), and inhibit the progression of AML [128]. Ethanol-induced mtDNA damage impaired the phagocytosis of alveolar macrophages (Figure 3) [129]. Therefore, we speculate that endogenous mtDNA accumulation in the cytoplasm may also result in death or reduced phagocytosis of macrophages following radiotherapy and chemotherapy.

#### 4.2.3. Impact of Swallowed mtDNA on Macrophage Biology

MtDNA contains the remnants of bacterial nucleic acid sequences and an inflammatory unmethylated CpG motif. In addition, many other nucleic acid species are produced, such as long double-stranded RNA, uncapped mRNAs, and RNA–DNA hybrids, accompanied by mtDNA transcription and replication [130]. Stressed mitochondria result in these nucleic acids entering the cytoplasm, where they promote immune cell activation and a pro-inflammatory response. The ability of mtDNA to drive greater production of type I interferons than nuclear DNA (nDNA) highlights the essential role of mtDNA in mediating innate immune responses.

After recognizing mtDNA, cyclic GMP-AMP synthase (cGAS) catalyzes the production of second messenger cyclic GMP-AMP (cGAMP), which then binds to STING, activating either the tank-binding kinase 1 (TBK1)-Interferon regulatory factor 3 (IRF3) or nuclear factor-κB (NF-κB)-IRF3 signal axis, which initiates the transcription of type I interferons and other cytokines [131,132]. Motwani et al. reviewed, in detail, how the cGAS-STING pathway-activated cytokines and phosphorylated organelle localization [133]. 

Radiotherapy-induced type I interferons, which depend on the cGAS-dependent cytoplasmic DNA sensor pathway, are necessary for anti-tumor immunity. Deng et al. also emphasized that the STING signal plays a vital role in the efficacy of radiotherapy combined with immunotherapy, driving an adaptive immune response to radiation [134]. Therefore, due to mtDNA stimulating the activation of the cGAS-STING pathway and causing the release of type I interferons, mtDNA plays a key role in activating immunological effects following conventional radiotherapy and chemotherapy. Although mtDNA is richer in the cytoplasm of cancer cells following irradiation compared to untreated cells, cancer cells rarely produce and secrete type I interferons. TAMs that swallowed irradiated tumor cells released mtDNA and further produced interferon-α/β in an autocrine manner [135]. However, the true nature of the tumor-derived molecular-activating STING phosphorylation of macrophages remains unclear. In addition to the mtDNA in macrophages directly activating the STING pathway, double-stranded DNA (dsDNA) directly activates cGAS in cancer cells; this activation produces cGAMP, which is then transferred to DCs and macrophages through gap junctions, which activates the STING pathway [136]. 

In addition to the classic STING pathway, mtDNA also activates toll-like receptors (TLRs), including TLR3, TLR7, TLR8, and TLR9, in monocytes. MtDNA and toll-like receptor interactions induce type I interferon production depending on the adaptor protein, myeloid differentiation factor88 (MyD88), and junction protein-containing TIR domain (TRIF) [137,138]. TLR9, which is expressed in the endoplasmic reticulum (ER) of a variety of immune cells, can recognize the DNA hypomethylated CpG motif in lysosomal compartments [74]. MtDNA transported to lysosomal compartments interacts with TLR9, initiating MyD88 dependent immune responses, activating mitogen-activated protein kinases (MAPK) and NF-κB and subsequently promoting the production of various pro-inflammatory cytokines in tumors [139]. In addition, TLR9 also produces type I interferons through IRF7 in DCs and other immune cells. High-mobility group protein B1 (HMGB1) can form a complex with mtDNA and bind to TLR9 to enhance inflammatory signals [140]. The TLR9-dependent inflammation induced by mtDNA not only occurs in 

TME but also in many TLR9-induced systemic immune system diseases, such as systemic lupus erythematosus, rheumatoid arthritis, atherosclerosis, myocarditis, dilated cardiomyopathy, acute liver injury, and nonalcoholic steatohepatitis [140,141].

Other PRRs, such as nod-like receptors (NLRs), can also recognize mtDNA. The release of both mROS and mtDNA into the cytoplasm enhances the activity of pyrin domain-containing protein 3 (NLRP3) [82]. In macrophages, mtDNA and ox-mtDNA were observed to activate NLRP3, exposing NLRP3′s pyrin domain, which binds to apoptosis-associated spot protein (ASC) to recruit the effector molecule pro-caspase-1, thereby forming NLRP3 inflammasomes [142]. NLRP3 was found to trigger self-cleavage and activation of caspase-1, transforming pro-IL-1β and pro-IL-18 into mature forms [143]. In the process of macrophage apoptosis, mtDNA is released and binds to NLRP3. Kenichi Shimada et al. further found that NLRP3 is inclined to bind to ox-mtDNA [144]. Recent studies have obtained similar results showing that species that bind to NLRP3 are newly synthesized ox-mtDNA, which activates NLRP3 and stimulates the secretion of IL-1β [77]. However, the activation of NLRP3 depends on either DNA or several other different signals [145]. From the perspective of inflammation, free ox-mtDNA can activate innate immunity, remodel the tumor microenvironment, and transform cold tumors into hot tumors (Table 3).

## 5. Anti-Cancer Therapeutic Opportunities That Involve MtDNA

MtDNA also has direct impacts on tumor progression and treatment. Glioblastoma and astrocytes can release mtDNA-carrying microvesicles [148]. MDVs are small vesicles that can carry mitochondrial proteins to other organelles. Additionally, mtDNA in MDVs was shown to stimulate the production of cellular pro-inflammatory cytokines, further enhancing LPS-induced inflammation [149,150]. However, Nicolas Rabas et al. found that tumor cells can promote the release of mtDNA-containing extracellular vesicles (EVs), allowing EVs to maintain their own growth and activating TLR9 to drive endosomal trafficking of the membrane type 1 matrix metalloproteinase (MT1-MMP), which promotes the destruction of the mammary basement membrane, thereby promoting breast cancer invasiveness [151]. MtDNA-containing exosomes can also exchange substances between tumor cells. For example, in a triple-negative breast cancer (TNBC) model, exosomes containing mutant mtDNA from drug-resistant TNBC cells can be transferred to drug-sensitive cells, increasing their chemoresistance [152]. Similarly, up-regulating mitochondrial Lon resulted in oxidized mtDNA, the carriage of mtDNA by EVs, and the release and secretion of PD-L1 into the cytoplasm. Oxidized mtDNA and EVs induced macrophages to produce IFN and IL-6 to up-regulate PD-L1 and IDO-1, which further weakened T-cell immunity. MtDNA release was found to trigger the expression of IFN-γ/PD-L1 in oral squamous cell carcinoma, regulate the immunosuppressive effect of macrophages, and inhibit the activation of T cells [153]. Depleting cytoplasmic mtDNA via deoxyribonuclease I (DNase I) or blocking the TLR9 pathway via the TLR9 antagonist significantly decreased the recruitment and polarization of TAMs in overexpressed dynamin-related protein-1 (Drp1) hepatoma cells [146]. Therefore, mtDNA-carrying exosomes seem to have a negative impact on anti-tumor effects.

Injured mtDNA can also promote anti-tumor therapy. For example, MRF-induced mtDNA stress can polarize TAM to the pro-inflammatory phenotype, thereby promoting IL-12, IL-6, TNF- α, IL-18, and IL-1 β in primary and distant tumors and serum, significantly inhibiting the growth of primary and distal tumors [109]. The universality and diversity of mtDNA’s influences on TAM suggest that we should carry out personalized design treatments and adopt different combined treatment methods to maximize treatment benefits, thus improving the survival rates and prognoses of tumor patients.

## 6. Conclusions

Cunning cancer cells not only disguise themselves to avoid being recognized by immune cells but also provoke macrophages by secreting cytokines TGF-β, CSF 1, and VEGF and tumor-derived exosomes (TDEs) or acidifying the TME generated by the special metabolic patterns of cancer cells [154,155]. Macrophages with both anti-tumor and pro-tumor phenotype cells play an important role in promoting tumor development [156,157]. With a deeper understanding of the immune escape of tumor cells and macrophage regulation, the plasticity of macrophages could be used to remodel the TME and promote anti-tumor therapy. Combination radiotherapy or chemotherapy with immunotherapy could achieve effective and long-term antitumor effects by regulating the phenotypic functions of macrophages.

Radiotherapy and chemotherapy can cause cancer cells to experience oxidative stress, nucleic acid damage, and release. Distinct from genomic DNA, each type of mtDNA derived from tumor cells or inherent in macrophages displays distinct biology. MtDNA and its downstream inflammatory products play different roles in the functions and phenotypes of macrophages in different tumor types. Even in the same tumor type, due to the complexity of TME, mtDNA has a certain impact on the survival, phenotypes, and functions of macrophages. Therefore, we speculate that mtDNA could also be used as a target for regulating TAM to promote anti-tumor effects, such as facilitating macrophage recruitment and survival to the tumor, domesticating the TAMs phenotype to the “M1-like” mode, and regulating macrophage phagocytosis.

In addition to acting as a DAMP that induces a monocyte inflammatory response to promote the transformation of TME from a cold to hot tumor, the mtDNA in neutrophil NETs can induce immunosuppression. Therefore, the effect of mtDNA on specific immune cell proportions and functions should be recognized. 

According to the proportion of immune cells in the tumor microenvironment, promoting or inhibiting the accumulation of mtDNA may be more beneficial to the remodeling of the TME. The content of mtDNA in the microenvironment is thus controlled according to mtDNA release and regulation—for example, (1) by promoting mtDNA damage in tumor cells. Promoting the accumulation of ROS in mitochondria and inducing the instability of mtDNA can directly cause cancer cell death and release antigens such as mtDNA. Specifically targeting mitochondrial RNA polymerase in tumors can inhibit mitochondrial transcription, thereby preventing ovarian and rectal cancer progression [158]. Inducing cell cycle arrest in the G2-M phase and increasing oxidative stress can, moreover, promote the sensitivity of glioblastoma cells to radiotherapy [159]. The second factor involves (2) regulating the release efficiency of mtDNA. Inhibiting the expression of TFAM can promote the sensitivity of tumor cell mitochondria to oxidative stress. Alternatively, the direct release of mtDNA can be regulated by a variety of MPTPs, VDAC, hexokinase, BAX, and BAK using a variety of protein agonists or inhibitors. Experiments have confirmed that mtDNA dependent on the release of BAK/BAX can activate cGAS-STING to induce interferon-β (IFN-β). When BAK/BAX is knocked out, mtDNA release is inhibited, IFN-β production is reduced, and the antitumor effect was attenuated [160,161]. 

In normal human cells, VDAC overexpression also triggers a type I interferon response induced by mtDNA release, ultimately leading to systemic lupus erythematosus [81]. Therefore, we can use these mechanisms to promote tumor mtDNA release, promote tumor sensitivity to radiotherapy and chemotherapy, and facilitate immune responses. The third factor is (3) regulating the accumulation of mtDNA in the TME. Promoting the necroptosis of tumor cells can enhance the effects of radiotherapy. For example, the inhibition of caspase-8 can significantly enhance activation of the ZBP1-MLKL pathway in tumor cells after radiotherapy and promote mitochondrial damage to release mtDNA [79]. The efficiency of intracellular DNA degradation can be regulated by inhibiting mitophagy or reducing the accumulation of DNase, thereby controlling the content of mtDNA in the TME. Mitophagy has been shown to suppress the innate immune response and promote immune escape by inhibiting activation of the NLRP3 inflammasome and the production of type I interferons [162,163,164]. The amount of free mtDNA can also be affected by the dose of radiotherapy and chemotherapy. Compared with high-dose radiotherapy, low-dose multiple radiotherapy can inhibit TREX1 expression and mtDNA degradation and promote a more favorable inflammatory environment [165]. Using these mechanisms can promote the accumulation of mtDNA in and out of cells to regulate immune responses. The fourth factor is (4) amplifying mtDNA-induced inflammatory responses. For example, inhibiting the CASP9 signaling pathway can help tumor cells recognize their own mtDNA, efficiently secrete IFN-I after radiotherapy, promote the cross-presentation of DCs, and enhance the anti-tumor immunity mediated by CD8^+^ T cells [166]. Theoretically, inhibiting the CASP9 signaling pathway in combination with other drugs that promote endogenous cell apoptosis could also inhibit tumor therapy resistance and enhance anti-tumor immune responses. Finally, the overall immunological effect promotes lasting and effective antitumor effects. The fifth factor involves (5) inhibiting M2-subtype TAM infiltration in the TME. At present, most mtDNA studies on TAM focus on the phenotypic transition of TAM. Whether mtDNA can directly inhibit the infiltration or function of M2 macrophages deserves to be further explored. Of course, based on the mtDNA-targeted anti-tumor therapy, not only the effect of anti-tumor therapy but also the biological safety of treatment needs to be considered to avoid producing an inflammatory factor storm, which can damage normal tissues and organs. However, the diversity of mtDNA regulation and TAMs and the signaling between them can be fine-tuned by the host metabolism and immune homeostasis, making this pathway challenging as a new therapeutic strategy.

In summary, understanding the mechanisms of mtDNA in remodeling the TME and the interactions between mtDNA and macrophages and properly designing individual treatment schemes could complement traditional radiotherapy, chemotherapy, or targeted therapy to achieve more efficient anti-tumor effects and improve the prognosis of patients.

## Figures and Tables

**Figure 1 cancers-14-01452-f001:**
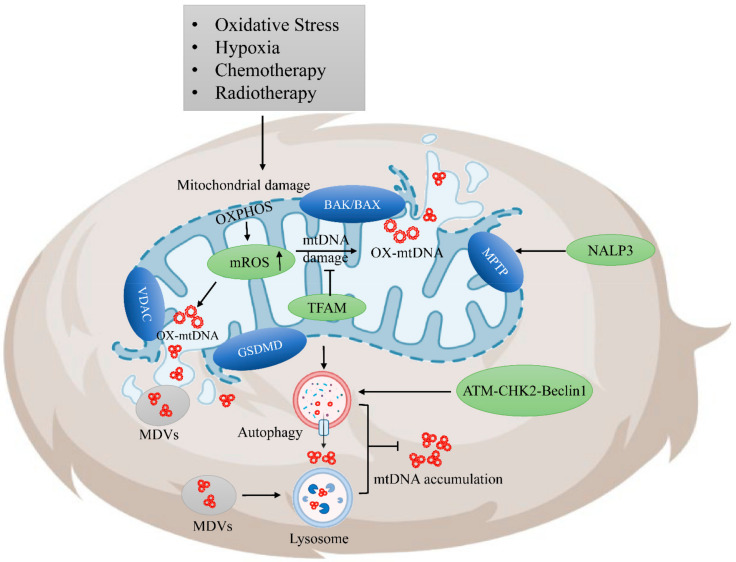
Regulation of tumor-derived mitochondrial DNA release. Damaged mitochondrial DNA (mtDNA) is released into the cytoplasm through mitochondrial membrane channels: When cells are exposed to exogenous radiotherapy, chemotherapy, or oxidative stress, reactive oxygen species (ROS) in the mitochondria increases dramatically, and damaged mtDNA is released into the cytoplasm in the form of ox-mtDNA fragments or mitochondrial-derived vesicles (MDVs) transport. The release of mtDNA depends on the mitochondrial permeability transition pore (MPTP) in the inner membrane of the mitochondria, which is partly formed by voltage-dependent anion channel (VDAC) oligomers in the outer membrane. Activation of the apoptotic protein Bcl-2 homologous killer/Bcl-2-associated X protein (BAK/BAX) can induce destruction of the mitochondrial outer membrane network, resulting in the appearance of BAK/BAX pores. Gasdermin protein can induce ruptures in the mitochondrial membrane network and promote the release of mtDNA into the cytoplasm. Pyrin domain-containing protein 3 (NLRP3) can promote the formation of MPTP in mitochondria, which allows the release of mtDNA. Autophagy and apoptotic caspase cascades drive the rapid decomposition and clearance of apoptotic cells and reduce mtDNA accumulation. MtDNA regulation in the cytoplasm: Tumor cells can transport damaged mitochondria and mtDNA to lysosomes for degradation through autophagy. The ATM-CHK2 pathway is also an important pathway transduction protein in DNA damage repair responses (DDRs), promoting autophagy to maintain cell homeostasis. MDVs can directly deliver damaged mitochondria to lysosomes without relying on autophagy.

**Figure 2 cancers-14-01452-f002:**
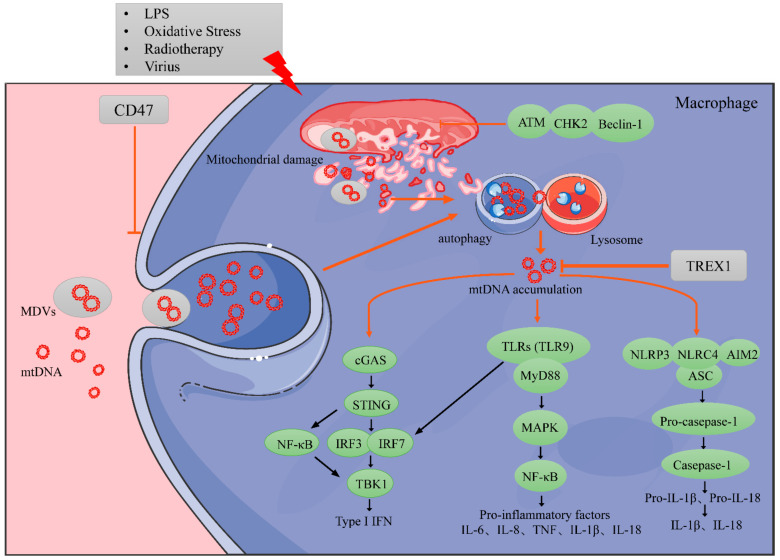
Source of mtDNA and immune responses in macrophages. MtDNA accumulation in macrophages: When tumors receive ionizing radiation, DNA-damaging drugs, and oxidative stress, the mitochondria of macrophages in the tumor microenvironments (TMEs) are damaged and release fragmented mtDNA. On the other hand, macrophages recognize the damaged mtDNA from tumor cells as exogenous foreign bodies and engulf them through endocytosis or MDVs. When activated, antigenic surface determinant protein OA3 (CD47) inhibits the phagocytosis of macrophages. The efficient fusion between phagosomes and lysosomes in macrophages results in rapid acidification of the phagosomal lumen and degradation of mtDNA. The three initial repair exonuclease 1 (TREX1), located in the cytoplasm of phagocytes, also degrades mtDNA in the cytoplasm. Inflammatory responses caused by mtDNA accumulation in macrophages: When mtDNA accumulates to a certain threshold, it can act as the damage-associated molecular pattern (DAMP) of cells and trigger inflammatory responses as well as cause innate immunity, including activating the cGAS-STING pathway, inducing the transcription and secretion of type I interferons, and promoting the production of a variety of pro-inflammatory cytokines when recognized by toll-like receptors (TLRs) and activated inflammatory bodies.

**Figure 3 cancers-14-01452-f003:**
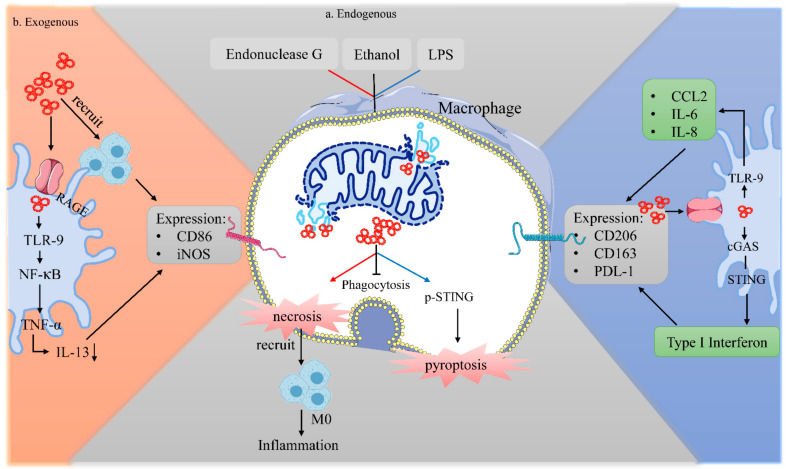
The biological effects of endogenous and exogenous mtDNA on macrophages. Endogenous mtDNA can regulate the death of macrophages: Endonuclease G enzymatically decomposes the mtDNA of macrophages in the traumatic tissue into fragments, and the accumulated mtDNA induces macrophage necroptosis, which further promotes the inflammatory response of surrounding naïve macrophages and forms an inflammatory microenvironment. The accumulation of damaged mtDNA induced by ethanol impairs the phagocytosis of macrophages. In the LPS-induced mitochondrial injury model, the amount of mt-DNA in the cytoplasm of macrophages increases, triggering stimulator of interferon genes (STING) phosphorylation and macrophage pyroptosis, inflammation, and oxidative stress. Peripheral mtDNA can promote the recruitment of macrophages to the TME and promote the polarization of macrophages to a proinflammatory phenotype. Inflammatory reaction products induced by mtDNA also have significant effects on the phenotypes of macrophages.

**Table 1 cancers-14-01452-t001:** Characteristics of M1 and M2 (M2a, M2b, M2c, and M2d) subtype macrophage.

	Stimuli	Markers	Secret	Function	Mechanisms	Ref
M1	IFNγ LPS GM-CSF	CD40 CD86 CD80 CD68 MHC-II IL1R TLR2 TLR4 SOCS3	TNF-α IL-1α IL-1β IL-6 IL-12 IL-18 IL-23 CCL4 CCL5 CCL8 CCL10 CCL11 CXCL9	Inflammation, tissue damage, pathogen clearance, inhibit cancer invasion and metastasis, enhance the metastatic potential of ovarian cancer cells	Enhances antigen presentation, activates Th1 cells, and secretes pro-inflammatory cytokines, release TNF-α to active NF-κB signaling pathway	[38,39,40]
M2a	IL-4 IL-13 Fungal/helminth infection LPS IFN-ɣ	Ym-1 CD163 MHC-II SRs CD206 YM1a FIZZ1a ARG1a CD86 iNOS Fizz	CCL17 CCL22 CCL24 RELM-α TGF-β IGF NO IL-10	Inflammation, tissue damage, pathogen clearance, tissue proliferation and repair and fibrogenesis, involved in parasitic infections	TGF-β1 promotes matrix synthesis and remodeling, arginase-mediated hydrolysis of arginine that drives the production of ornithine to promote fibrogenesis, support efficient IFN-ɣ production in CD8^+^ T cells	[28,33,41,42,43,44,45,46]
M2b	IL-1R LPS IFN-ɣ IL-4	CD86 MHC-II IL-10 TNF IL-1β IL-6 MerTK SPHK1 LIGHT	IL-10 IL-1 IL-6 IL-10 TNF-α CCL1	Anti-inflammation, Minor tissue damage, Ameliorated myocardial ischemia/reperfusion injury, tissue remodeling, promote tumor development and infections	inhibited IFN-ɣ expression in CD4^+^T, active kinase of platelet-derived growth factor receptor of cardiac fibroblast, inhibit the immune and inflammatory response	[47,48,49,50,51,52,53]
M2c	IL-10 TGF-β Glucocorticoids IFN-ɣ IL-6	CD163 TLR-1 TLR-8 IL-10 TGF-β CD206 SLAM MerTK	IL-10 TGF-β CCR2	Anti-inflammation, tissue remodeling, phagocyte apoptotic cells	Reduce CD4^+^T cell activation and proliferation, TGF-β1 promotes matrix synthesis and remodeling, high expression of MerTK	[45,54,55,56]
M2d	IL-6 TLR ligands A2R Adenosine	VEGF IL-10 TGF-β	IL-10 IL-12 TNF-α TGF-β CCL5 CXCL10 CXCL16	Tissue repair, Angiogenesis, promote cancer growth and metastasis	Regulate integrin (avb3) receptors and Src-PI3K-YAP signaling to promote angiogenic activity, secrete anti-inflammatory cytokines and suppress T-cell immunity	[37,57,58,59,60,61]

**Table 2 cancers-14-01452-t002:** Effects of mtDNA on different immune cell populations.

Cell Populations	Protein Expression	Function	Anti-Tumor Effects	Ref
Dendritic Cells	(↑) CD86 (↑) CD83 (↑) HLA-DQ (↑) TNF-α (↑) IFN-β (↑) IL-3 (↓) CXCR4 (↓) CXCR3 (↑) CCR7	(↑) activation (↑) maturation (↑) migration	Anti-tumor immunity	[78,95,96,104,105]
Tumor-associated neutrophil	(↑) myeloperoxidase (MPO) (↑) type I IFN	(↑) recruitment (↑) activation induces NETs	accelerated progression, facilitated metastatic seeding and progression, poor prognosis	[99,100,102,103]
B cell	(↑) type I IFN	-	-	[106]
CD4^+^T cell	(↑) IL-10	(↑) activation	Anti-tumor immunity	[104,107]
CD8^+^T cell	-	(↑) proliferation	Anti-tumor immunity	[78]

↑ upregulation, ↓ downregulation.

**Table 3 cancers-14-01452-t003:** The effects of mtDNA on macrophage.

mtDNA Source	Mechanisms	Effects	Polarization	Anti-Tumor Effects	Ref
Endogenous	(↑) P-STING (↑) TNF-α	Inflammation, oxidative stress, pyroptosis	(↑) M1	-	[110,126]
	mtDNA accumulation	Necroptosis, inflammatory in surrounding naive macrophages	(↑) M1	-	[127]
	Ethanol-induced mtDNA exosome release	(↓) Phagocytosis		-	[129]
	cGAS-STING, TLRs-MyD88, NLRs-ASC	Inflammation, (↑) Type I IFN, IL-6, IL-8, TNF, IL-1β, IL-18	(↑) M1	Remodeling the TME	[77,132,133,137,144]
Exogenous	-	TAM recruitment, (↑) CD86, (↓) CD206	(↑) M1	Inhibit progression and growth of distant tumors in pancreatic cancer	[108,109]
	TLR-9 induce CCL2, IL-6, IL-8 production	TAM infiltration, MDSCs recruitment	(↑) M2	Promote progression in HCC, Promote epithelial-mesenchymal transition and metastasis in gastric cancer	[113,146,147]
	STING induce IFN-I production	(↑) PD-L1,IL-10; (↓) IL-1b	(↑) M2	-	[118]
	TLR9- NF-κB and cGAS-STING induce TNF-α production	Inflammation, (↓) IL-13	(↑) M1	Modify the inflammatory microenvironment, anti-tumor	[110,111,112]

↑ upregulation, ↓ downregulation.

## Data Availability

Not applicable.

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
