# Peer review of "Mitochondrial DNA on Tumor-Associated Macrophages Polarization and Immunity"

_cancers, 2022, doi:10.3390/cancers14061452_

Round 1
Reviewer 1 Report
The authors give a very complete review of the double edge sword function that mitochondrial DNA has in immune regulation in inflammatory diseases and cancer.
As the authors show clearly targeting this pathway as a novel therapeutic strategy seems difficult if not impossible as outcome of its signaling is fine tuned by the metabolic and immune homeostasis of the host.
A more detailed discussion discussing how to selectively target mitochondrial DNA release and signaling in healthy and disease would add great value to the current review.
Reviewer 2 Report
The literature review by Zhu et al. describes the impact of mitochondrial DNA on functions of tumor-associated macrophages. The review is interesting and comprehensive. I have a few suggestions to the authors.
- Make sure that all molecules described in the review are correctly identified. In one instance, adenosine triphosphate is labeled as Arg-1.
-
The authors suggest that induction of M1 macrophages can be used for treating cancer. Why is inhibition of M2 macrophages is not a viable therapeutic approach?
- I suggest that the authors consider adding a separate section focused on anti-cancer therapeutic opportunities that involve mitochondrial DNA.
- I recommend a careful revision of the text to improve readability.
Reviewer 3 Report
In their work, Guo et al. discussed the role of mtDNA on the regulation of TAMs polarization and immunity. In general, paper is interesting and describes mechanisms that are of general interests in immunotherapy field. However, it has to be improved to improve its scientific standard.
Major points
In most paragraphs, authors refer to the review articles, not original articles that indeed described discussed mechanisms or processes. It has to be corrected. Moreover, there are multiple theses or information that are not supported by appropriate references.
Authors should add a table that will compare M1, M2 (M2a,b,c, and M2d) - their markers, role in the regulation of immune response, mechanisms etc.
Figures should be improved. In the current version there are unclear and are not very legible.
Authors should add a table summarizing the influence of mtDNA on TAMs
Authors should add a table comparing the effect of mtDNA on different immune cell population (see Lines 195-196)
Authors should more deeply described the role of TAMs in the regulation of anti-tumor immunity, since it is the key topic of their paper.
Authors did not describe the exosome-packed mtDNA and their role in anti-tumor immunity.
A recent important reference demonstrating the role of acetoacetate in protecting from mitochondrial dysfunction is missing (Adam C. et al. Nat Comm 2021). Moreover, authors should more deeply describe mechanisms that result in the increased production of mtDNA in TME as well as describe the mechanisms that protects TAMs from the mtDNA-induced stress.
Minor comments
Line 38 - remove "positively"
Line 48 - cite relevant paper
Line 52 - there are much more different mechanisms that result in the increased mutational rate of mtDNA compared to gDNA. It has to be discussed more deeply.
Line 57 - remove word "systematically" as it may suggest that authors performed a systematic review of the literature which is not true.
Lines 62-70 - what references support these theses?
Line 100 - "adenosine triphosphate (Arg-1)" - do authors mean ATP or arginase 1 (Arg-1)? Please correct this.
Lines 104-105 - remove duplication of this sentence.
Line 181 - use "tumor microenvironment" instead of "tumor immunity environment"
Round 2
Reviewer 3 Report
Guo et al. revised their manuscript according to Reviewer's comments.
I must congratulate the authors for their efforts. Table 1 and Table 2 are great addition to the paper. Figures are also significantly improved.
I have just some minor points that will improve their manuscript:
1. Line 71 - please add reference. Moreover, it would be better to add the information that the percentage of TAMs vary between different types of tumors
2. Please include recent review article when describing ARG1 in TAMs (f.e. PMID: 32499785)
